# Precision Medicine in Pancreatic Ductal Adenocarcinoma: The Impact of Targeted Therapies on Survival of Patients Harboring Actionable Mutations

**DOI:** 10.3390/biomedicines11092569

**Published:** 2023-09-19

**Authors:** Anthony Tarabay, Alice Boileve, Cristina Smolenschi, Leony Antoun, Marine Valery, Alina Fuerea, Audrey Perret, Pascal Burtin, Simona Cosconea, Hichem Belkhodja, David Malka, Valérie Boige, Antoine Hollebecque, Michel Ducreux

**Affiliations:** 1Gustave Roussy, Département de Médecine, 94805 Villejuif, France; alice.boileve@gustaveroussy.fr (A.B.); cristina.smolenschi@gustaveroussy.fr (C.S.); leony.antoun@gustaveroussy.fr (L.A.); marine.valery@gustaveroussy.fr (M.V.); alinacristina.fuerea@gustaveroussy.fr (A.F.); audrey.perret@gustaveroussy.fr (A.P.); pascal.burtin@gustaveroussy.fr (P.B.); simona.cosconea@gustaveroussy.fr (S.C.); hichem.belkhodja@gustaveroussy.fr (H.B.); david.malka@gustaveroussy.fr (D.M.); valerie.boige@gustaveroussy.fr (V.B.); michel.ducreux@gustaveroussy.fr (M.D.); 2Gustave Roussy, Département d’Innovation Thérapeutique et d’Essais Précoces, 94805 Villejuif, France; antoine.hollebecque@gusatveroussy.fr; 3Faculty of Medicine, Université Paris Saclay, 91400 Orsay, France

**Keywords:** PDAC, precision medicine, gene alterations, molecular profiling, molecular targeted therapies

## Abstract

Background: Pancreatic ductal adenocarcinoma (PDAC) is the third leading cause of death by cancer worldwide. Mostly diagnosed with locally advanced or metastatic disease, patients lack treatment options. Gene alterations (GAs) are frequently observed in PDAC, some of which are considered for molecular targeted therapies (MTTs), with potential clinical benefits and improved outcomes. However, the applicability of molecular profiling (MP) for precision medicine in PDAC remains to be demonstrated. Methods: We conducted a retrospective analysis of all patients, aged ≥18 years with histologically confirmed PDAC, who underwent tumor MP between 2010 and 2020 in our institution as part of personalized medicine trials. The primary study endpoint was overall survival (OS), and (minimal follow-up was 6 months after MP). Results: Of 115 eligible patients, MP was successful in 102 patients (89%). KRAS mutations were the most frequent GAs, mostly G12D. Based on ESCAT classification, actionable GAs were found in 29 patients (28%), involving mainly *BRCA1* or *BRCA2* (5 (18%)), *HER2* (5 (18%)), *MTAP* (5 (18%)), and *FGFR* (3 (11%)). Only 12 of these 29 patients (41%, or 10% of the whole population) received MTTs, with a median progression-free survival of 1.6 months. Median OS was 19 months in patients with actionable GAs treated with MTTs (*n* = 12 (11.8%)), 14 months in patients with actionable GAs treated with standard therapies (*n* = 17 (16.7%)), and 17 months in patients without actionable GAs treated with standard therapies (*n* = 73 (71.5%); *p* = 0.26). The absence of liver metastases was associated with better OS (HR = 0.471, *p* = 0.01). The highest OS following MTT was observed in patients with BRCA mutations treated with olaparib. Interpretation: Actionable GAs were found in more than a quarter of patients with advanced PDAC. Overall, targeting actionable GAs with MTTs was not associated with improved OS in this retrospective study with limited patient numbers. However, selected GA/MTT combinations (e.g., BRCA mutations/olaparib) were associated with a better outcome.

## 1. Introduction

Pancreatic ductal adenocarcinoma (PDAC) is the most common type of pancreatic cancer, especially in Asian countries but also in Europe and the United States [1]. With a slight male predominance [2], this cancer is the third leading cause of cancer-related death in Western countries [1,3]. Only 20–30% of patients are diagnosed with resectable or borderline resectable tumors. In these patients, even with the adjuvant FOLFIRINOX, less than 30% are cured [4]. Patients who exhibit either a locally advanced or a metastatic disease at diagnosis cannot be treated with surgery. In locally advanced disease, a small number of patients can be treated with surgery after neoadjuvant treatment, including combined chemotherapy with or without chemoradiation, and a small percentage are cured after these intense treatments. Nevertheless, most patients die from their disease, especially in metastatic cases. The standard-of-care in first-line metastatic disease relies on a combination of 5-fluorouracil (5FU) plus irinotecan plus oxaliplatin (the FOLFIRINOX regimen) [4] or a combination of gemcitabine and nab-paclitaxel [5,6]. With these regimens, median overall survival (OS) is close to 10 months [4,5]. At progression, a second line of treatment is proposed to ~60% of patients. In patients receiving gemcitabine-based chemotherapy as the first line, oxaliplatin-based second-line treatment has shown conflicting results; therefore, the most evidence-based second-line option is the use of nanoliposomal irinotecan combined with 5FU and folinic acid (NALIRI regimen) [7].

Molecular profiling (MP) of PDAC discovered several key gene alterations (GAs) involved in carcinogenesis mainly in four genes: *KRAS*, *TP53*, *SMAD4*, and *CDKN2A* [8]. Except for the *KRAS* G12C mutation, which is found in less than 2% of PDAC cases, these GAs currently have no molecular target therapies (MTTs). Nevertheless, in around 25% of cases, actionable GAs have been found at low frequencies [9,10,11,12], including genes involved in the DNA damage response and repair (DDR); some specific pathways (BRCA1, BRCA2, PALB2); the AKT, PI3K, FGFR, NTRK, MYC, MET, and NOTCH pathways; amplification of ERBB2; and mutations in RNF43, mTOR, PDGFR, ROS1, KIT, BRAF [8,13].

However, very few MTTs have been proven to be effective, and only olaparib (POLO trial) has been approved as treatment in patients who have responded to a first-line of platinum-based chemotherapy, with locally advanced or metastatic pancreatic cancer carrying germline mutation of BRCA1 or BRCA2 [14].

In recent years, several prospective studies have been developed to evaluate the feasibility and efficacy of precision medicine using next generation sequencing (NGS) in patients with various solid tumors [15,16,17,18,19,20]. Of note, only 10–25% of patients in these studies received a specific therapy informed by MP [15,16,17,18,19,20], and fewer benefited from this approach. However, a recent prospective randomized precision medicine trial in patients with breast cancer reported that the use of MTT improved progression-free survival (PFS), but only when GAs were classified as level I/II based on the ESMO Scale for Clinical Actionability of Molecular Targets (ESCAT) [18]. This was confirmed by data from the precision medicine program at our institution [21].

In PDAC patients specifically, results from several prospective and retrospective studies showed a benefit in terms of overall response rates, PFS, and median overall survival (OS) in patients with an actionable GA who received the corresponding MTT, compared with those who had an actionable GA but not treated with MTT [6,10,22,23]. In their recently published retrospective study, Pishvaian and colleagues found a positive impact on OS of treatments targeting actionable mutations in PDAC. Their study showed better OS in patients harboring actionable Gas treated with MTT compared to those treated with standard therapies [10]. Nevertheless, whether a MP approach could improve treatment decision-making is yet to be demonstrated in patients with PDAC.

In this retrospective study, we analyzed the clinical outcomes of patients with PDAC from whom we had obtained their MP through our institutional precision medicine programs. We aimed to report the clinical applicability of precision medicine in PDAC and assess the impact of MTT in patients with actionable GAs.

## 2. Materials and Methods

### 2.1. Study Design

We conducted a retrospective analysis of all >18 year-old patients with histologically confirmed PDAC who underwent tumor MP between 2011 and 2020 in our institution as part of personalized medicine trials. We excluded patients with noncontributive, or when obtained using liquid biopsy only. Included patients had received at least one line of treatment at our institution.

Clinical characteristics, as well as treatment-related outcomes, were retrospectively collected using a hospital chart review (demographics, clinical characteristics at diagnosis, primitive lesion treatments, systemic treatments, and outcomes). The primary endpoint was OS, defined as time between diagnosis of metastatic disease and death or loss of follow-up. A minimal follow-up of 6 months was required after molecular profiling. Secondary endpoints were PFS for patients receiving MTTs.

This retrospective study complies with the French MR004 methodology regarding general data protection regulation for noninterventional retrospective health research (Délibération n° 2018-155 du 3 mai 2018) and was approved by our institutional review board (CSET N°2023-175), in compliance with the Helsinki declaration. All patients provided signed informed consent for genomic analyses as part of precision medicine trials.

### 2.2. Genomic Analyses

MOSCATO

The primary objective of both versions of the MOSCATO study was to evaluate the clinical benefit of with evaluation of outcomes following MTTs. This trial included patients from December 2011 until March 2016. The inclusion criteria [15] specified all patients with a histologically proven solid tumor (including PDAC), refractory or in locoregional or metastatic relapse considered as not curable with conventional treatments, were eligible. Patients then underwent a biopsy of an accessible tumor site (primary tumor or metastasis) according to the local recommendations of our institution.

A conventional histological evaluation was performed first to determine the percentage of tumor cells. Samples showing more than 30% tumor cells were selected for whole molecular analysis. In samples with 10–30% tumor cells, only targeted sequencing was performed.

DNA and RNA extractions were performed using the Qiagen method (RNeasy/Dneasy) at the genomic functional unit (GFU). Array comparative genomic hybridization (aCGH) was performed on DNA for samples having a quantity of DNA greater than 1 μg. High-level amplifications (log2ratio < 0.89) and genomic loss (log2ratio < −0.28) were determined by the bio informaticians of the GFU.

Separately, 700 ng of DNA was used for screening of somatic mutations, covering more than 5000 mutations described in the COSMIC database. The Ion AmpliSeq Cancer Panel (CP1), covering 190 amplicons in 40 cancer genes (ThermoFisher Scientific, Les Ulis, France), was used between May and November 2012. From December 2012 to September 2013, the panel was expanded (Ion AmpliSeq Cancer Hotspot Panel v2) to include 207 amplicons in 50 cancer genes (ThermoFisher Scientific). Finally, starting September 2013, the last panel (MOSC3) covered 75 cancer genes using Ion AmpliSeq custom design, combining 1218 amplicons with the CHP2 panel.

RAGNAR trial

As part of the RAGNAR phase II trial on the efficacy and toxicity profile of erdafitinib in patients with solid tumor with an alteration of the FGFR gene, eligible patients benefited first from a molecular screening for an alteration in the FGFR gene. Archived tumor tissue or fresh samples were sent to the central laboratory for testing. After DNA extraction, molecular analysis was carried out with NGS based on the FoundationOne^®^CDx platform, allowing for the detection of mutations in 324 genes as well as the status of microsatellite instability and tumor mutational burden.

STARTRK trial

As part of the STARTRK phase II basket trial, which evaluated entrectinib in the treatment of patients with locally advanced or metastatic solid tumors carrying the NTRK1/2/3, ROS1, or ALK gene rearrangements, eligible patients benefited from a first phase of molecular screening, based on the FoundationOne^®^CDx platform, allowing the detection of mutations in 324 genes, as well as the determination of and status.

### 2.3. Treatment Decision

For all patients, results were discussed during a weekly multidisciplinary precision medicine tumor board. Results were reported in the medical file of each patient. Actionability of GAs was classified according to ESMO Actionability for Molecular Targets (ESCAT) guidelines [24]. Considerations for MTT based on GAs relied on variant annotation databases (OncoKB, CIViC, My Cancer Genome, and the literature) as well as European Medicines Agency (EMA) approval, temporary authorization of use (ATU) of the drug and phase I clinical trials available in our institution. In the absence of an actionable GA, the patient received the physician’s standard treatment of choice.

Hence, three groups of patients were defined: group A included patients with actionable GAs who received MTT; group B included patients with actionable GAs who did not receive MTT but were treated with standard treatment; and group C included patients without actionable GAs who received the physician’s standard treatment choice.

### 2.4. Statistical Analysis

The groups were compared with the chi-squared test or Fisher’s exact test for binary variables and with the Mann–Whitney nonparametric test for continuous variables. Survival was calculated from the date of onset of metastases. Multivariate analysis was performed with a Cox model. The effect of administering targeted therapy was studied, considering targeted therapy as a time-dependent variable. Other studied variables were the presence of hepatic or lung metastases, the number of metastatic sites, and sex. All statistical analyses were performed using Rstudio software (version 4.3.0).

## 3. Results

Out of 115 eligible patients, 102 patients got MP (exclusion causes were: biopsy contraindication for coagulopathy (n = 3), cancelled biopsy for intercurrent disease (n = 2), noncontributive sample (n = 5), and no accessible lesion (n = 3) in patients without available tumor tissue) (Figure 1).

Patients’ characteristics are presented in Table 1. With a slight male predominance (55.9%), the mean age at diagnosis was 59 years [37; 75]. More than half of the patients were metastatic at diagnosis (61.8%), with mainly hepatic metastases (76%). MP was performed 2 years after diagnosis most of the time. At that time, 98% of patients presented with metastatic disease, and 75% of them had already received two or more lines of treatment.

After performing the molecular sequencing, 359 abnormalities in 94 different genes were observed in the whole population (Figure 2). Most of these abnormalities were mutations (n = 232, 64.6%), but there were also gene amplifications (n = 48, 13.4%), gains (n = 31, 8.6%), and deletions (n = 48, 13.4%) (Figure 3). The most frequently altered gene was KRAS (found in 87 patients out of 102, 85.3%) followed by TP53 (in 77 patients, 75.5%) and CDKN2A (in 23 patients, 22.5%). Regarding KRAS, the most frequent mutation was G12D in 51% of cases, followed by G12V and G12R. The G12C mutation was found only in four patients. Regarding other types of GAs, loss of CDKN2A and CDKN2B were found in 12 and 11 patients, respectively (11.8% and 10.8%). HER2 was amplified in four patients (3.9%).

Twenty-nine patients (28.4%) exhibited a potentially actionable molecular alteration according to ESCAT classification at time of inclusion (ESCAT I-IV). The most common actionable mutations were amplification of HER2 and mutations in DNA repair genes (BRCA2 and PALB2), n = 5 for both (Figure 4).

Of these 29 patients, 12 patients received MTT. Reasons for not receiving MTT for the other 17 patients are detailed in Table 2. MTT was usually received as the second line of treatment (median: 2; [1, 4]).

The MTTs were olaparib (in three patients with BRCA mutations), trastuzumab-based treatment regimen (in three patients with HER2 amplification), NOTCH 1 inhibitor (in one patient), anti-MEK (in one patient harboring a BRAF mutation), AMG510 (in one patient with a KRAS G12C mutation), erdafitinib (in one patient with an FGFR alteration), AG-270 (in one patient with an MTAP deletion), and AZD6738 (an ATR inhibitor used in one patient with a BRIP1 alteration).

The median duration of response to MTT was 1.55 months (95%CI: 1.1—not reached (NR)). Despite the low sample size, patients with BRCA2 mutation seemed to have longer PFS following olaparib treatment, reaching 13.1 months in one patient, while a second patient was still on treatment at the time of the statistical analysis (follow-up of 7 months) (Figure 5).

Out of the 17 patients with actionable GA but not treated with MTTs, only 9 patients could receive treatment after MP was performed, with PFS not exceeding 4 months. Treatments were chosen depending on physician’s preferences and the prior lines (Figure 6).

Regarding OS, there was no significant difference between the three groups, with a median OS of 19 months in group A, 14 months in group B, and 17 months in group C (*p* = 0.4889) (Figure 7). Multivariate analysis using the Cox model with time-dependent variables revealed better OS of patients without liver metastases compared with those with liver metastases at the time of molecular profiling with HR = 0.471 (*p* = 0.01).

## 4. Discussion

Here, we report our real-life data of the clinical applicability of precision medicine in a tertiary center for the therapeutic management of PDAC in routine clinical practice. We show that is feasible in PDAC, but only 45% of patients were treated with MTT when actionable GAs were found.

In our study, we found a comparable number of molecular profiles to previous studies [8,9,10,11] with the most common mutations in *KRAS* (85.3% of which 51% were G12D), followed by *TP53* (67%) and *CDKN2A* (10%). With 28.4% of patients having a potentially actionable mutation based on ESCAT classification, our results are similar to that exhibited in Pishvaian et al. [10], in which 26% of patients had potentially actionable mutations [10]. Despite the limited number of patients actually treated with MTT, this is similar to that already described in large scale precision medicine trials for solid tumors [15,16,25]. The proportion of patients receiving MTT was low due to several reasons. First, accessibility to treatment is difficult, especially accessibility to early phase I trials. Indeed, these trials are frequently restricted to a preselected population. Second, it is generally only performed after several treatment lines have failed, when the general health status of the patient is poorer, which limits the potential clinical benefit from MTT. Performing at diagnosis or in early treatment might circumvent the rapid decrease in health status frequently observed in patients with PDAC. Third, the notion of mutation actionability is dynamic, and new drugs might allow previously nonactionable molecular alteration to become actionable. The key mutation in PDAC is *KRAS*, which is notoriously resistant to drugs; however, *KRAS* inhibitors are indeed emerging, with the recent FDA approvals in non-small cell lung cancers of two *KRAS* G12C-targeting drugs, sotorasib [26] and adagrasib [27]. In the small subset of patients with PDAC (n = 12), results of sotorasib were encouraging, with one partial response and nine stable diseases. Despite these encouraging results, their impact is limited given the small proportion of patients with PDAC associated with the G12C mutation in *KRAS* (1–2%) [27]. With ~40% of PDAC cases harboring a *KRAS* G12D mutation [28], the pending trials of long-awaited *KRAS* G12D inhibitors should be more promising.

*BRAF* targeting therapy has changed over time and varies depending on the underlying pathology. Currently, anti-*BRAF* anti-*MEK* double blockade is the standard of care. In our study, the patient with a *BRAF* mutation was treated with only a *MEK* inhibitor, which can explain the poor PFS observed with this treatment and, consequently, the unimproved OS. In PDAC, only few case reports concluded for a potential benefit of the combination therapies [29,30].

Therefore, unlike other studies highlighting a positive impact of therapies targeting actionable mutations on OS, our study shows no difference in survival curves in the three defined groups.

The absence of liver metastases appears as a good prognostic factor for overall survival, with a decrease in the risk of death of 53% (*p* = 0.01), independent of group. These results are concordant with the analysis of prognostic factors made in recent randomized studies of pancreatic cancer [4].

Our study has several limitations. First, this is a retrospective and monocentric study. Second, sample sizes were limited, with few patients actually treated with MTT. However, this is attributable to PDAC treatment recommendations as few MTT are approved. MTTs are available for very rare mutations, such as for high tumor mutational burden (TMB); however, the molecular landscape of PDAC is dominated by *KRAS* mutations, the inhibitors of which are pending. Third, it was performed on both primary and metastatic tumor cells, which might induce heterogeneity in the results. However, this heterogeneity reflects real-life clinical practice, and available tissues for NGS analyses (specific biopsy or archived tissue) were chosen as part of routine therapeutic management with the least risk for the patient. Finally, we did not address impact of liquid biopsy: a simple, noninvasive, effective method that is rapidly expanding [31] and might be of interest for PDAC patients. Further exploration of the potential impact and the clinical utility of liquid biopsy will be needed for further PDAC treatment studies.

## 5. Conclusions

Our study reported on real-life precision medicine with outcomes regarding routine in clinical practice. A potential actionable mutation was found in 28% of patients in advanced PDAC, of which 44% received an MMT; however, there was no improvement in OS.

## Figures and Tables

**Figure 1 biomedicines-11-02569-f001:**
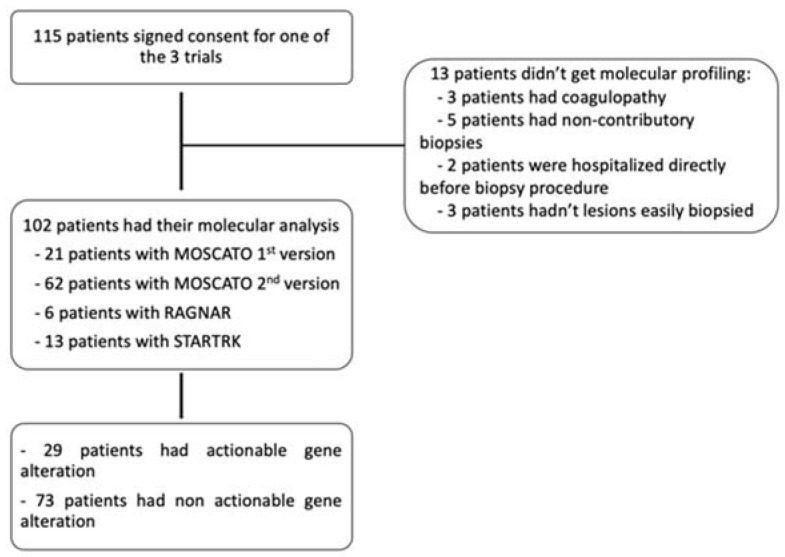
Patient selection algorithm.

**Figure 2 biomedicines-11-02569-f002:**
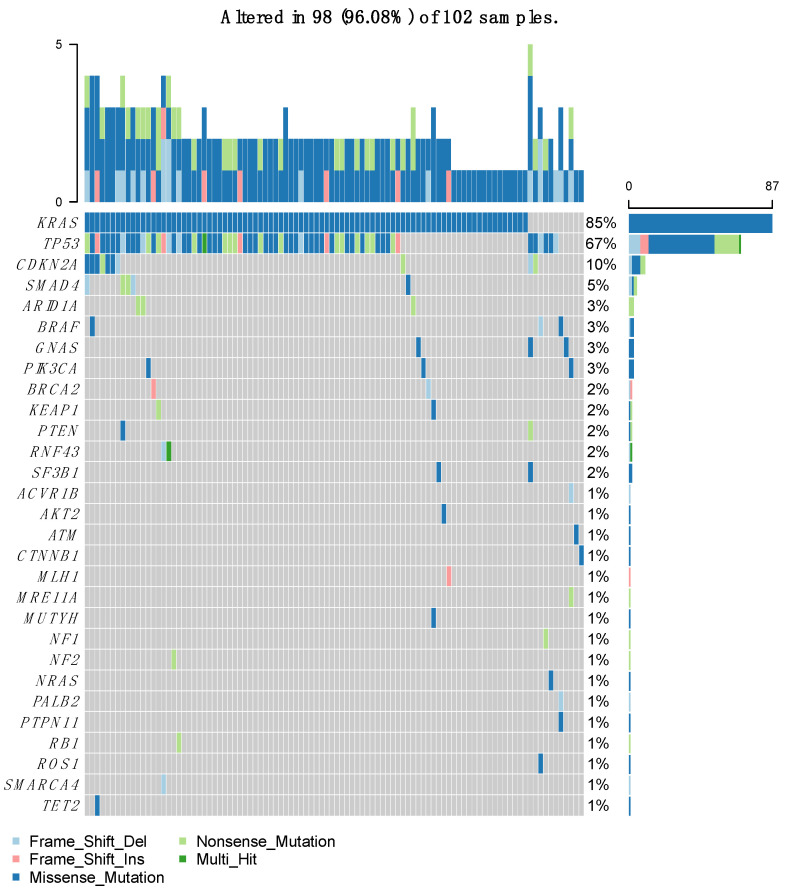
Summary of most frequent gene mutations found in 102 patients; 85% of patients had a *KRAS* missense mutation.

**Figure 3 biomedicines-11-02569-f003:**
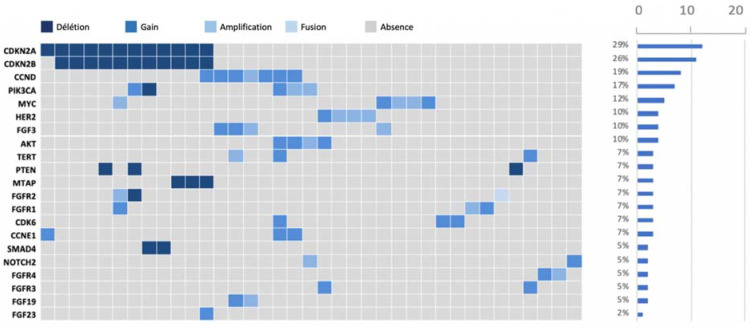
Summary of most frequent nonmutation gene alterations; the most affected genes were CDKN2A and CDKN2B.

**Figure 4 biomedicines-11-02569-f004:**
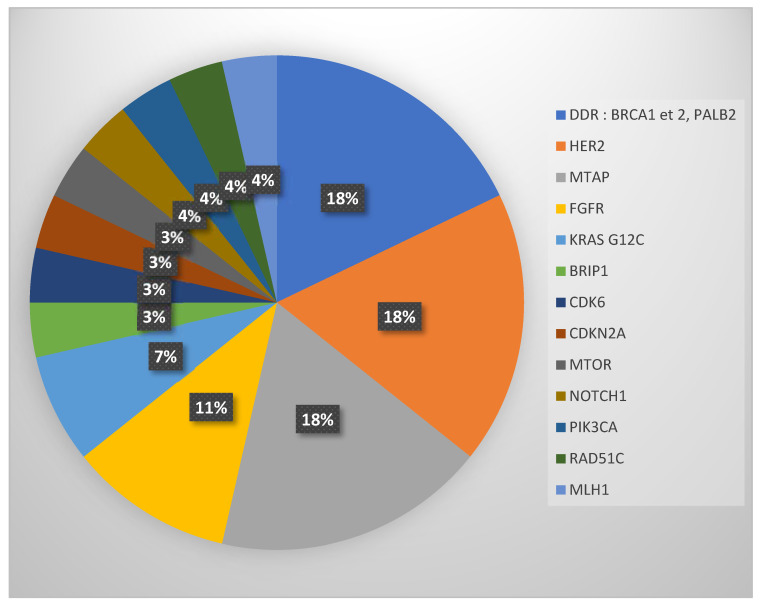
Diagram of gene alterations considered as “actionable” at the time of molecular profiling. These alterations mostly implicated genes of the DDR pathway, HER2, MTAP, and FGFR.

**Figure 5 biomedicines-11-02569-f005:**
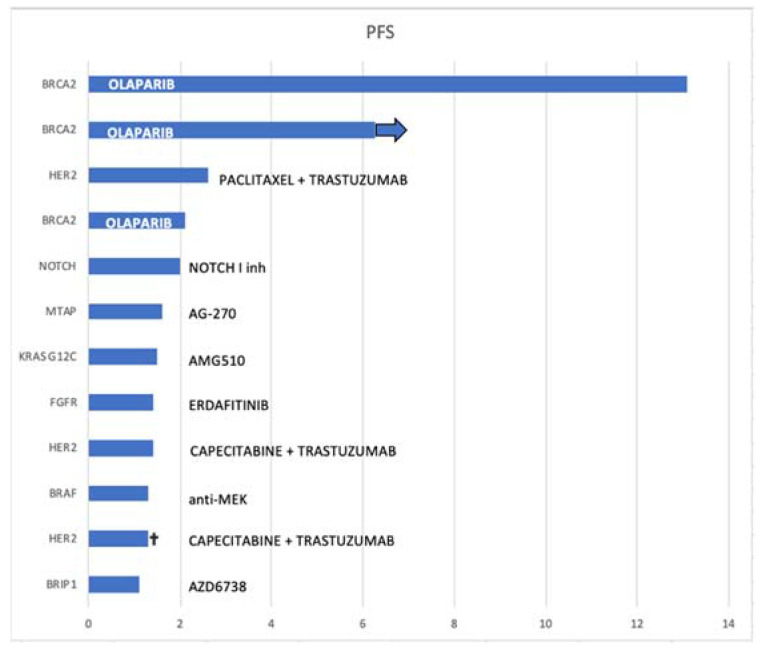
Progression-free survival (PFS) of 12 patients following molecular target therapy out of 29 harboring actionable gene mutations (
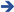
 ongoing treatment; **†** dead). Better PFS is seen in patients with BRCA2 mutations that were treated with olaparib.

**Figure 6 biomedicines-11-02569-f006:**
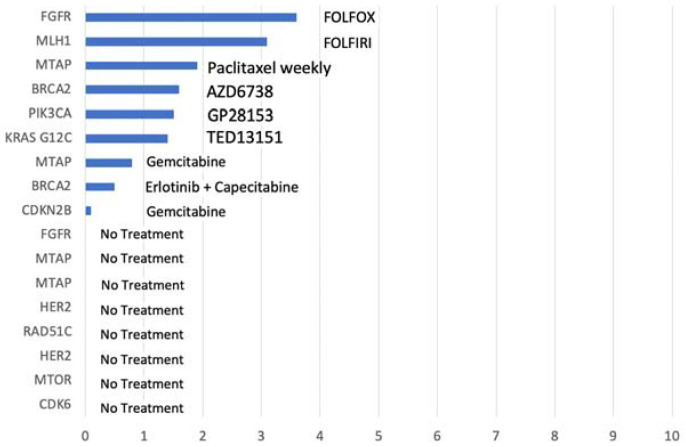
PFS of 17 patients harboring actionable GA but not treated with MTTs; treatment at first progression directly after performing MP.

**Figure 7 biomedicines-11-02569-f007:**
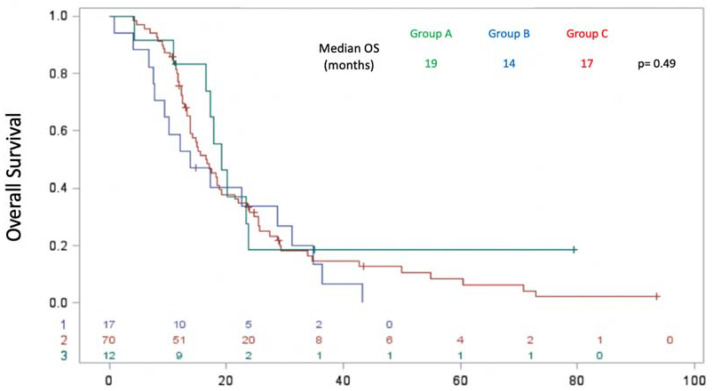
Overall survival for each group: A (patients with actionable GAs and treated with MTT), B (patients with actionable GAs but not treated with MTT), and C (patients with no actionable GAs).

**Table 1 biomedicines-11-02569-t001:** Patient characteristics.

		Total		Group A		Group B		Group C		*p*
Sex	n (%)	102		12		17		73		
	Male	57	(55.9)	6	(50.0)	13	(76.5)	38	(52.1)	0.1716
	Female	45	(44.1)	6	(50.0)	4	(23.5)	35	(47.9)	
Age at diagnosis	n	102		12		17		73		
	Mean	57.8		58.3		58.3		57.9		
	Median	59		60		60		59		
	Min; Max	30; 75		30; 68		40; 75		30; 75		
Age at molecular profiling	n	102		12		17		73		
	Mean (sd)	59.8	(10.7)	58.1	(11.5)	63.0	(11.3)	59.4	(10.5)	0.3322
	Median	61 (52; 68)		63 (52; 67)		64 (54; 72)		60 (51; 68)		
	(Q1; Q3)									
	Min; Max	31; 78		31; 69		41; 77		32; 78		
Stage at molecular profiling	n (%)	102		12		17		73		
	Stage I–III	2	(2.0)					2	(2.7)	0.6668
	Stage IV	100	(98.0)	12	(100.0)	17	(100.0)	71	(97.3)	
Nb of prior treatments	n (%)	102		12		17		73		
	≤ 1 line	27	(26.5)	4	(33.3)	5	(29.4)	18	(24.7)	0.7260
	>1 line	75	(73.5)	8	(66.7)	12	(70.6)	55	(75.3)	
Nb of metastatic sites	n (%)	100		12		17		71		
	1 site	78	(78.0)	9	(75.0)	12	(70.6)	57	(80.3)	0.8340
	> 1 site	22	(22.0)	3	(25.0)	5	(29.4)	14	(19.7)	
Liver metastasis	n (%)	100		12		17		71		
	No	24	(24.0)	3	(25.0)	3	(17.6)	18	(25.4)	0.9314
	Yes	76	(76.0)	9	(75.0)	14	(82.4)	53	(74.6)	
Lung metastasis	n (%)	100		12		17		71		
	No	85	(85.0)	9	(75.0)	15	(88.2)	61	(85.9)	
	Yes	15	(15.0)	3	(25.0)	2	(11.8)	10	(14.1)	
Peritoneal carcinosis	n (%)	100		12		17		71		
	No	88	(88.0)	10	(83.3)	15	(88.2)	63	(88.7)	
	Yes	12	(12.0)	2	(16.7)	2	(11.8)	8	(11.3)	

*p* = *p* value of Chi-square test or Fisher’s exact test for qualitative variables. *p* = *p* value of Wilcoxon rank-sum test for quantitative variables.

**Table 2 biomedicines-11-02569-t002:** Reasons for not receiving molecular target therapies (MTTs).

Reasons for Not Receiving Molecular Target Therapies (MTTs)	Number (%)
Poor performance status at progression	9 (53.0)
Patients having some of the exclusion criteria of the trial offering the MTT	4 (23.6)
Patients refusing consent	2 (11.7)
Other causes	2 (11.7)

## Data Availability

Not applicable.

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
