# Peer review of "Precision Medicine in Pancreatic Ductal Adenocarcinoma: The Impact of Targeted Therapies on Survival of Patients Harboring Actionable Mutations"

_biomedicines, 2023, doi:10.3390/biomedicines11092569_

Round 1

Reviewer 1 Report

In this article, Tarabay et al, analysed retrospectivelly the efficiency/applicability of molecular profiling for precision medecine in pancreatic ductal adenocarcinoma patients. The cohorte analysed is heterogeneous but conclusions could be made on 102 patients.

The topic is very important and complex to set up. The relatively low incidence of PDAC makes such study difficult. I am interested by this retrospective study, that shows that the « apparent » failure of targeted therapies in PDAC without patient stratification should not preclude from success with well selected therapies. I am favorable for the publication of this article pending two comments:

-          In the abstract and throughout the text, it should be stressed that they analysed usually single cases for each MP/therapy except for BRCA2/olaparib, correspon DOI: 10.1056/NEJMoa1903387 ;

-          Figure 5 : add the data for patients with actionable GAs treated with standard treatment.

Minor

-          l247 correct »benefitted »

-          l274 add ref

Author Response

Dear Editor

Thank you for taking the time to review the manuscript entitled "Precision medicine in pancreatic ductal adenocarcinoma: the impact of targeted therapies on survival of patients harboring actionable mutations".

i reviewed all your suggestions and concluded as follow:

  • Figure 5: we only presented PFS of 12 patients following molecular target therapy. The main purpose was to show that when we treat patients with targeted therapies according to their GA, and that are not considered as standard therapies, these treatments can offer good PFS as shown in patients treated with olaparib. Patients treated with standard therapies could have received multiple lines of treatment. So which line should we select to evaluate its PFS? Especially  that Molecular Profiling was performed at different time for each patient.
  • The line 274: this paragraph is presenting our study results, that's why we didn't put reference.

Thank you again for your help and waiting for your suggestions.

Regards

Anthony TARABAY

Reviewer 2 Report

In the present article titled “Precision medicine in pancreatic ductal adenocarcinoma: the impact of targeted therapies on survival of patients harboring actionable mutations” the authors conducted a retrospective analysis on patients with PDAC, who underwent tumor molecular profiling for precision medicine. The objectives of this study are the evaluation of overall survival and progression free survival in patients with actionable alteration receiving molecular target therapies.

This study has limitations (among which the sample size) but, despite that, the authors obtained encouraging results in terms of PFS in some cases and highest OS only in patients with BRCA mutations and treated with olaparib.

Overall, the manuscript is well written, even if some minor stylistic revisions should be made.

I think this article is acceptable for publication in Biomedicines.

Author Response

Dear Reviewer

Thank you for your feedback.

Regards

Anthony Tarabay